# Soil Physicochemical Changes as Modulated by Treated Wastewater after Medium-and Long-Term Irrigations: A Case Study from Tunisia

Sinda Bekir [1],*[ID], Rahma Inès Zoghlami [1,2],†[ID], Khaoula Boudabbous [3],†, Mohamed Naceur Khelil [4], Mohammed Moussa [1][ID], Rim Ghrib [4], Oumaima Nahdi [3], Emna Trabelsi [3] and Habib Bousnina [1]

1   Laboratory of Eremology and Combating Desertification, Arid Regions Institute, University of Gabes, Medenine 4119, Tunisia
2   National Institut of Agronomic Research, Laboratory of Agricultural Sciences and Techniques, Rue HédiKarray, Cp1004 Menzah 1, Ariana 2049, Tunisia
3   Horticultural Sciences Laboratory, LR13AGR01, National Agronomic Institute of Tunisia, University of Carthage, Tunis-Mahragene 1082, Tunisia
4   National Research Institute of Rural Engineering, Water and Forestry, University of Carthage, LR 7 16INRGREF02, LR Valorization of Unconventional Waters, 17 Rue HédiKarray, 8 BP No. 10, Ariana 2080, Tunisia
*   Correspondence: bekirsinda11@gmail.com; Tel.: +216-5244-9075
†   These authors contributed equally to this work.

**Abstract:** Treated wastewater (TWW) is considered as an alternative for agricultural irrigation. The aim of this study was to understand the medium- and long-term effects of TWW on soil physicochemical parameters. Two perimeters ($P_1$ and $P_2$)receiving TWW for 38 and 20 years, respectively, in Tunisiawere selected for study. In each perimeter, two water types were adopted: TWW and groundwater (GW). Soil physicochemical traits (pH, EC, and concentrations of $Na^+$, $K^+$, $Ca^{2+}$, and $Mg^{2+}$) were measured up to 100 cm, and three indexes were calculated: sodium adsorption ratio (SAR), cation ratio of structural stability (CROSS), and cation exchange capacity (CEC). Overall, all soil parameters were significantly affected in the irrigation area using GW. However, in the case of TWW, only the pH was found to be affected, increasing by 4.7% from $P_1$ to $P_2$. Moreover, compared to GW, TWW enhanced the soil salinity by 127%, particularly at deeper subsoils. More interestingly, the results revealed an accumulation of $Mg^{2+}$, $Ca^{2+}$, and $K^+$ and a depletion of $Na^+$ at the soil surface. Notably, TWW showed the lowest CROSS and SAR indexes, indicating the benefits of applying TWW even after long-term use in improving soil physicochemical parameters such as sodicity and structural stability. Our results provide valuable information for decision-makers to use wastewater in irrigated marginal soils.

**Keywords:** water reuse; sandy soil; physicochemical parameters; irrigation period; depth

## 1. Introduction

Due to the world's rising fresh water scarcity, the reuse of treated wastewater (TWW) for irrigation has been proposed as a substitute for using fresh water, particularly in arid and semiarid regions such as Middle East and North Africa (MENA). In these regions, irrigation water shortage has madetreated wastewater an attractive source of water for sustainable agriculture in order to better preserve fresh water for human consumption [1]. In this regard, Thebo et al. [2] stated that over 20 million hectares of land wasirrigated globally using reclaimed wastewater and that this number would significantly rise over the next few decades as water stress increases. In Tunisia, the reuse of treated wastewater in agriculture is an old practice. It dates back to 1965 with the creation of the first perimeter of Soukra (Governorate of Ariana). Since the last century, according to the National Sanitation Utility (NSU), the numbers of perimeters and treatment plants have continued to increase

to reach 122 plants treating 284 mm$^3$, which should serve 8435 hectares of agricultural area. In addition, TWW reuse for agriculture offers some attractive environmental and socioeconomic benefits. Compared to soil that has been irrigated with fresh water, TWW reuse can enrich soils with nutrients such as N, P, and K [3]. Thus, the richness of TWW in mainly nitrogenous fertilizing elements should lead to a reduction in input of mineral fertilizers. Consequently, it allows an economic gain for farmers [4]. Additionally, due to the high levels of BOD present, TWW is inhabited by a wide range of bacteria that promote organic matter decomposition and preserve soil fertility. Thus, irrigation managed by TWW could lead to better productivity [5].

Despite significant benefits, recycled water may deteriorate soil health in terms of increased salinity and sodicity [6]. Indeed, the effect of TWW on soil salinity has been the subject of several studies, but the results remain controversial [7]. According to many works, the reuse of TWW increases soil salinity, especially in sandy soils [8]. According to Changati et al. [9], water from wastewater irrigation enhanced the exchangeable sodium concentration (Na$^+$) and exchangeable sodium cation percentage (ESP) because of higher electrical conductivity, total dissolved solids, and major ion concentrations. However, Ahmed and El-hedek [10] found a decrease in the soil EC and pH. Such results could depend on the variability of soil characteristics. The main factor affecting the management of TWW irrigation is the interaction between the total mass of salts in the soil, salinity of irrigation water (characterized by electrical conductivity, EC), and the ratio of sodium and other cations in the solution (represented by the sodium absorption ratio, SAR) [11]. Recognizing the role of K$^+$ and Mg$^{2+}$, Rengasamy and Marchuk [12] proposed a cation ratio of structural stability (CROSS) alternative to SAR. The CROSS was developed to reflect the different dispersive powers of Na$^+$ and K$^+$ and the different flocculating powers of Ca$^{2+}$ and Mg$^{2+}$. In this context, Moruga'n-Coronado et al. [13] showed a decrease in aggregate stability for soils irrigated with TWW due to the high Na$^+$ content. More interestingly, the effect of TWW on soil fertility might be influenced by the application period. In fact, numerous studies have documented increasing saline levels in the soil as a result of continuous irrigation using recycled water. A large number of researchers have similarly observed long-term salinity impacts of TWW in terms of EC, Na$^+$, and SAR [14]. While various studies [15,16] have demonstrated an increase in soil salinity as a result of recycled water irrigation, the phenomenon is highly variable depending on the soil parameters, water quality, and application period of TWW.

In Tunisia, there is consensus in the scientific literature about the impact of wastewater irrigation on the physical and chemical properties of soil. However, there is little information on the long-term impact of wastewater irrigation in sandy soil texture in semiarid climatic conditions. Consequently, the potential effect of TWW on soil properties in the medium and long-term requires additional research, especially in developing countries such as Tunisia. To contribute to the current understanding of soil fertility, two perimeters with differing TWW application length were studied in sandy soil texture. Different chemicals, soil properties, and sodicity and structural stability indexes were investigated in a region northeast of Tunisia. The aim of this study was to (i) assess the effect of TWW on soil physicochemical parameters after medium- and long-term application; (ii) highlight the relationships between CEC, SAR, CROSS indexes, and soil exchangeable cations; and (iii) provide additional information to farmers about irrigation management.

## 2. Materials and Methods

### 2.1. Environmental Background of the Study Area

This study was conducted at two sites corresponding to irrigation perimeters in the northeast of Tunisia, namely, Oued Souhil (P$_1$: 40G 40′9G 09′) and Beni Khiar (P$_2$: 40G 57′9G 40′) (Figure 1).

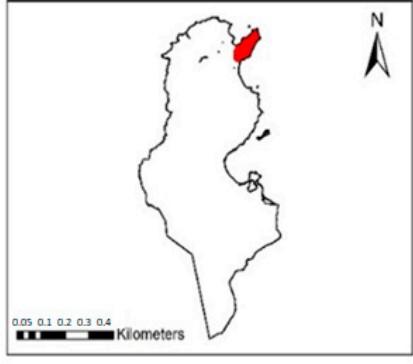

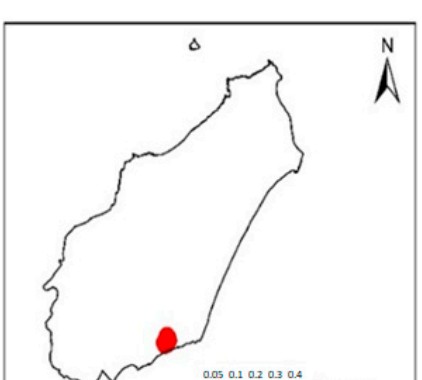

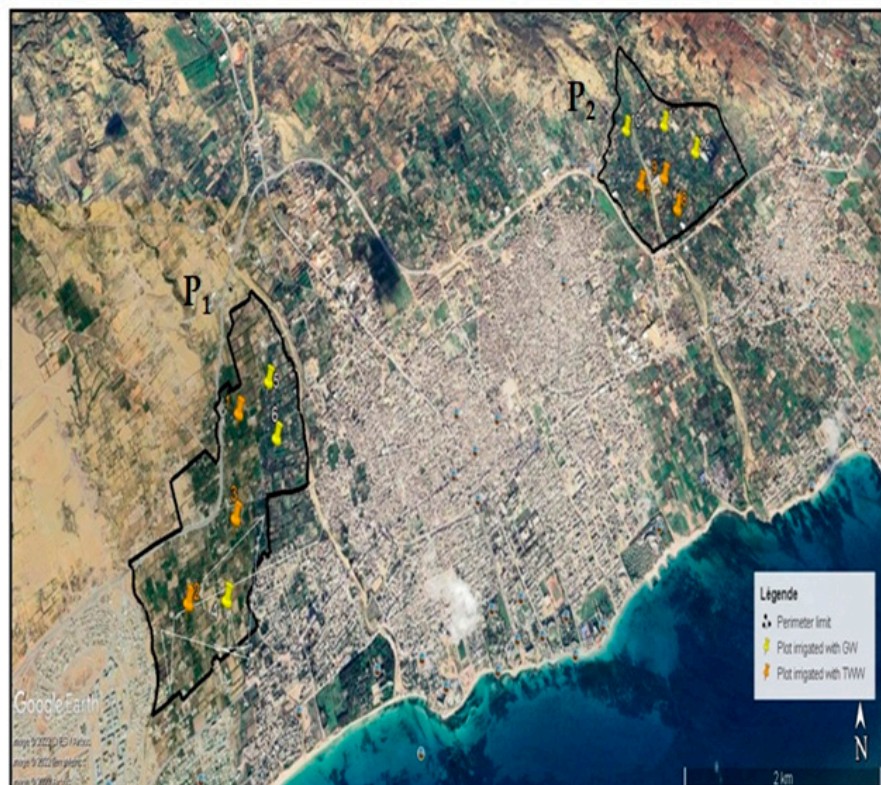

**Figure 1.** Map of the study areas.

The region of Nabeul is a coastal plain with a low extension. The geological outcrops of the region have ages ranging from the upper Miocene to the Quaternary [17]. According to the Regional Commissariat for Agricultural Development of Nabeul (RCAD Nabeul, 2020), this region is drained by a relatively dense hydrographic system. It is a series of exoreic water courses parallel totemporary flow and whose catchment basins are of limited extension. The two studied perimeters, $P_1$ and $P_2$, are crossed by Oued Souhil and Oued El Kebir, respectively. These two oueds are among the most important oueds of the Nabeul region and cover 43.8 and 22.1 km$^2$ of surface, respectively (Table 1).

**Table 1.** Physical characteristics of catchment basins El Kebir and Souhil(RDAC Nabeul, 2020).

| Catchment Basin | Surface (km$^2$) | Length (km) | Altitude (m) | Input (mm$^3$ year$^{-1}$) |
|---|---|---|---|---|
| El Kebir | 43.8 | 15.2 | 118 | 2.69 |
| Souhil | 22.1 | 12.5 | 104 | 1.35 |

For each of the two sites, $P_1$ and $P_2$, two sources of water were used: treated wastewater (TWW) and groundwater (GW) (RCAD Nabeul, 2020). The $P_1$ was irrigated for 38 years, while $P_2$was irrigated for 20 years. The groundwater used was part of the large coastal aquifer system of Nabeul–Hammamet, which extends over an area of 100 km$^2$. This aquifer is characterized by low thickness of the reservoir rock varying from 1 to 3 m, high salinity with more than 3.5 dS m$^{-1}$, and pH of 8.08. For more than 50 years, it has been overexploited, which has caused intrusion of the saline wedge along the coastal zone. The inventory of water points shows that the water table of Nabeul–Hammamet is exploited by 3930 surface wells producing about 15,106 m$^3$ of water. The second irrigated water source (TWW) was collected downstream of the Nabeul wastewater treatment plant, which receives domestic and industrial effluents. It is characterized by an average basic pH value of 7.7, and its EC reaches 4 dS m$^{-1}$ (Table 2). Table 2 summarizes the physicochemical

parameters. During the experiment, the mean annual temperature and rainfall registered were 19°C and 400 mm, respectively (RCAD Nabeul 2020).

**Table 2.** Mean values of chemical properties of treated wastewater (TWW) generated by the wastewater treatment plant of Nabeul and groundwater (GW) from 2016 to2020 (RDAC Nabeul 2020).

| Parameter | TWW | GW | Standars * |
|:---:|:---:|:---:|:---:|
| pH | 7.7 | 8.08 | 6.5–8.5 |
| EC (dS m$^{-1}$) | 4 | 3.5 | 7 |
| Cl$^-$ (mg L$^{-1}$) | 705 | 390 | 2000 |
| Na$^+$ (mg L$^{-1}$) | 516 | 590 | - |
| K$^+$ (mg L$^{-1}$) | 41.5 | 25 | - |
| Ca$^{2+}$ (mg L$^{-1}$) | 25 | 12 | - |
| Mg$^{2+}$ (mg L$^{-1}$) | 30 | 3 | - |
| SS (mg L$^{-1}$) | 17 | 4.3 | 30 |
| COD (mg L$^{-1}$) | 56 | 19 | 90 |
| BOD$_5$ (mg L$^{-1}$) | 30 | 4.27 | 30 |

* Tunisian standards for wastewater reuse (NT 106.03).

The soil texture at the two perimeters was sandy in all depths surveyed. Table 3 illustrates the average proportions of sand, clay, and silt of the two perimeters (P$_1$ and P$_2$) in each depth until 100cm.

**Table 3.** Soil texture in each depth of perimeters P$_1$ and P$_2$.

| Depth (cm) | Sand (%) | Clay (%) | Silt (%) | Soil Texture |
|:---:|:---:|:---:|:---:|:---:|
| 0–20 | 79.5 ± 1 | 12.5 ± 1.0 | 8.0 ± 1.0 | Sandy |
| 20–40 | 79.0 ± 0.5 | 13.5 ± 0.5 | 7.5 ± 0.5 | Sandy |
| 40–60 | 78.5 ± 0.3 | 14.5 ± 0.3 | 7.0 ± 0.3 | Sandy |
| 60–80 | 79.0 ± 0.5 | 13.0 ± 0.5 | 8.0 ± 0.5 | Sandy |
| 80–100 | 79.0 ± 0.5 | 14.0 ± 0.5 | 7.0 ± 0.5 | Sandy |

Percentage values are means of three replicates in each perimeter ± SD.

### 2.2. Experimental Design and Soil Sampling

At each irrigation perimeter (P$_1$ and P$_2$), the study adopted a completely randomized design with two main factors: two sources of irrigation water (GW and TWW) and five soil depths (0–20, 20–40, 40–60, 60–80, and 80–100 cm).

At each irrigated perimeter, three plots were selected for each irrigation source (TWW and GW). For each plot, sampling was carried out by taking soil from the five depths. In each soil depth, five cores were sampled and homogenized. Samples were collected using an auger for a total of 30 samples for each perimeter. The crop cultivated in the two perimeters was citrus. The soil samples were air-dried, sieved trough 2 mm mesh, kept in plastic bags, and transferred to the laboratory for pH, EC, and nutrient content analysis.

### 2.3. Element Analysis

The pH, electrical conductivity (EC),and concentrations of soluble cations Na$^+$, K$^+$, Ca$^{2+}$, and Mg$^{2+}$ were determined in soil pore water (saturated paste extract). To prepare this, 200 g of air-dried soil was used, and the soil pastes were left for 24 h to reach equilibrium. Subsequently, the vacuum extracts were collected [18].

Soil pH level was determined using the WTW inoLab 7110 pH meter. EC was measured using theinoLab 7110conductivity benchtop meter. The concentrations of soluble Ca$^{2+}$ and Mg$^{2+}$ were measured using the EDTA titration method [19], while the Na$^+$ and K$^+$ were determined using a flame photometer [20]. The nutrients K$^+$, Na$^+$, Ca$^{2+}$, and Mg$^{2+}$ were

used to calculate the sodium adsorption ratio (SAR) (Equation (1)) and the cations ratio of soil structural stability (CROSS) (Equation (2)).

$$\text{SAR} = \frac{\text{Na}}{\sqrt{(\text{Ca} + \text{Mg})/2}} \tag{1}$$

$$\text{CROSS} = \frac{(\text{Na} + 0.56\,\text{K})}{\sqrt{(\text{Ca} + 0.6\,\text{Mg})/2}} \tag{2}$$

Cation exchange capacity (CEC) was calculated using the sum of $Na^+$, $K^+$, $Ca^{2+}$, and $Mg^{2+}$ concentrations. All concentrations are expressed as millimole of charge$L^{-1}$ [12].

### 2.4. DATA Analysis

Collected data were subjected to the two-way analysis of variance (ANOVA) test, and differences between means were determined according to the Duncan significant difference test ($p < 0.05$) using SPSS software, version 16 S, to assess the differences in soil properties between the two irrigation practices under medium- and long-term irrigations in different soil depths. Pearson correlation coefficient analysis was used to examine the correlation between CEC, SAR, CROSS, and the different cations under each irrigation type.

## 3. Results

### 3.1. Soil pH

Our results showed that pH values depended significantly ($p < 0.05$) on the period of TWW application and the water type. In contrast to TWW, soil pH did not change with the period of irrigation using GW. Overall, we noted an increase in pH values when applying TWW, particularly in the oldest irrigated perimeter. In addition, regardless of the water type, data revealed a significant increase in pH, which remained alkaline through all soil depths. Using TWW, the pH changed across depths, with the values being 2.8% lower in $P_2$ than in $P_1$ at 0–60 cm (Table 4).

### 3.2. Soil Electrical Conductivity (EC)

The EC varied significantly according to water type and soil depth (D), while it was not significantly affected by the period of TWW application. In contrast, soil EC decreased by 29% after 38 years of using GW for irrigation. Regardless of the period of irrigation, in the case of TWW, the results showed an increase in EC by 34% from the top soil (0–20 cm) to deep soil (80–100 cm), while it was reverse in the case of GW, which indicated a decrease in EC by 29% from the top to deep soil layers (Table 4).

### 3.3. Cation Element Contents and Cation Exchange Capacity (CEC)

No significant effect was detected for cation element contents and the CEC according to the period of TWW application (Table 5). However, the results showed a decrease in CEC after 38 years of irrigation using GW. In addition, $Na^+$, $K^+$, and $Ca^{2+}$ contents decreased according to the irrigation period from 20 to 38 years of GW application by 20, 15, and 50%, respectively. Meanwhile, $Mg^{2+}$ concentration increased by 43% after 38 years of irrigation with GW. CEC and cation element contents remained dependent on the water irrigation type and depth (D).

**Table 4.** Changes in soil pH, CE, SAR, and CROSS according to soil depth and period of irrigation with GW (a) and TWW (b) in $P_1$ and $P_2$ areas.

| **(a) GW** | | | | | | | | |
|---|---|---|---|---|---|---|---|---|
| Period of irrigation with GW (IP) | pH | | EC (dS m$^{-1}$) | | SAR (mmole$^{0.5}$ L$^{-0.5}$) | | CROSS (mmole$^{0.5}$ L$^{-0.5}$) | |
| $P_1$ (38 years) | 8.19 | | 0.48 | | 1.37 | | 1.66 | |
| $P_2$ (20 years) | 8.23 | | 0.62 | | 1.51 | | 1.73 | |
| Depth (cm) (D) | $P_1$ | $P_2$ | $P_1$ | $P_2$ | $P_1$ | $P_2$ | $P_1$ | $P_2$ |
| 0–20 | 8.54 a ± 0.2 | 8.16 c ± 0.0 | 0.64 b ± 0.0 | 0.7 a ± 0.0 | 1.03 d ± 0.2 | 1.11 cd ± 0.0 | 1.43 c ± 0.2 | 1.66 b ± 0.0 |
| 20–40 | 8.17 c ± 0.0 | 8.07 d ± 0.0 | 0.48 e ± 0.0 | 0.69 a ± 0.0 | 1.49 b ± 0.0 | 1.4 b ± 0.0 | 1.91 a ± 0.0 | 1.59 b ± 0.0 |
| 40–60 | 8.33 b ± 0.0 | 8.09 d ± 0.0 | 0.38 g ± 0.0 | 0.64 b ± 0.0 | 1.74 a ± 0.1 | 1.81 a ± 0.0 | 1.97 a ± 0.2 | 1.92 a ± 0.0 |
| 60–80 | 8.08 d ± 0.0 | 8.36 b ± 0.0 | 0.48 e ± 0.0 | 0.58 c ± 0.0 | 1.41 b ± 0.0 | 1.48 b ± 0.0 | 1.7 b ± 0.0 | 1.61 b ± 0.0 |
| 80–100 | 7.87 e ± 0.0 | 8.48 a ± 0.0 | 0.42 f ± 0.0 | 0.53 d ± 0.0 | 1.21 c ± 0.0 | 1.76 a ± 0.1 | 1.3 c ± 0.0 | 1.88 a ± 0.1 |
| IP | ns | | * | | * | | * | |
| D | * | | * | | * | | * | |
| IP × D | * | | * | | * | | * | |
| **(b) TWW** | | | | | | | | |
| Period of irrigation with TWW (IP) | pH | | EC (dS m$^{-1}$) | | SAR (mmole$^{0.5}$L$^{-0.5}$) | | CROSS (mmole$^{0.5}$L$^{-0.5}$) | |
| $P_1$ (38 years) | 8.4 | | 1.3 | | 0.8 | | 1.1 | |
| $P_2$ (20 years) | 8.0 | | 1.2 | | 1.0 | | 1.3 | |
| Depth (cm) (D) | $P_1$ | $P_2$ | $P_1$ | $P_2$ | $P_1$ | $P_2$ | $P_1$ | $P_2$ |
| 0–20 | 8.3 ab ± 0.0 | 7.6 b ± 0.0 | 1.1 a ± 0.0 | 1.2 a ± 0.2 | 0.6 a ± 0.0 | 0.3 a ± 0.0 | 0.9 a ± 0.0 | 0.7 a ± 0.0 |
| 20–40 | 8.5 a ± 0.0 | 8.0 ab ± 0.0 | 1.3 a ± 0.0 | 0.8 a ± 0.0 | 0.8 a ± 0.0 | 0.8 a ± 0.0 | 1.1 a ± 0.0 | 1.0 a ± 0.1 |
| 40–60 | 8.6 a ± 0.0 | 8.2 ab ± 0.0 | 1.2 a ± 0.0 | 1.2 a ± 0.0 | 0.8 a ± 0.1 | 1.3 a ± 0.0 | 1.0 a ± 0.1 | 1.5 a ± 0.0 |
| 60–80 | 8.3 ab ± 0.0 | 8.1 ab ± 0.0 | 1.3 a ± 0.0 | 1.3 a ± 0.0 | 0.7 a ± 0.0 | 1.0 a ± 0.0 | 1.1 a ± 0.0 | 1.3 a ± 0.0 |
| 80–100 | 8.2 ab ± 0.0 | 8.2 ab ± 0.0 | 1.6 a ± 0.0 | 1.5 a ± 0.0 | 1.0 a ± 0.0 | 1.2 a ± 0.0 | 1.3 a ± 0.0 | 1.6 a ± 0.1 |
| IP | * | | ns | | ns | | ns | |
| D | ns | | ns | | ns | | ns | |
| IP × D | ns | | ns | | ns | | ns | |

GW: groundwater; TWW: treated wastewater; EC: electrical conductivity; SAR: sodium adsorption ratio; CROSS: cations ratio of soil structural stability; $P_1$: Oued Souhil area; $P_2$: Beni Khiar area. Significant differences between different levels foreach depth are indicated by lower case letters; ns: not significant, *: significant at $p < 0.05$.

**Table 5.** Changes in soil $Ca^{2+}$, $Mg^{2+}$, $K^+$, and $Na^+$ concentrations and CEC according to depth and irrigation period with GW (a) and TWW (b) in $P_1$ and $P_2$ areas.

**(a) GW**

| Irrigation period with GW (IP) | $Ca^{2+}$ (mmoleL$^{-1}$) | | $Mg^{2+}$(mmoleL$^{-1}$) | | $K^+$(mmoleL$^{-1}$) | | $Na^+$(mmoleL$^{-1}$) | | CEC (mmoleL$^{-1}$) | |
|---|---|---|---|---|---|---|---|---|---|---|
| $P_1$ (38 years) | 1.52 | | 0.72 | | 0.32 | | 1.4 | | 3.95 | |
| $P_2$ (20 years) | 2.29 | | 0.41 | | 0.37 | | 1.7 | | 4.79 | |
| Depth (cm) (D) | $P_1$ | $P_2$ | $P_1$ | $P_2$ | $P_1$ | $P_2$ | $P_1$ | $P_2$ | $P_1$ | $P_2$ |
| 0–20 | 2.04 d ± 0.0 | 3.64 a ± 0.0 | 0.68 d ± 0.0 | 0.11 g ± 0.0 | 0.69 b ± 0.0 | 0.31 a ± 0.0 | 1.2 h ± 0.2 | 1.52 def ± 0.0 | 4.62 b ± 0.2 | 6.58 a ± 0.1 |
| 20–40 | 1.14 g ± 0.0 | 2.5 b ± 0.2 | 0.57 e ± 0.0 | 0.23 f ± 0.0 | 0.47 c ± 0.0 | 0.35 d ± 0.0 | 1.38 g ± 0.0 | 1.63 cd ± 0.0 | 3.55 e ± 0.1 | 4.71 b ± 0.2 |
| 40–60 | 0.8 h ± 0.5 | 1.59 e ± 0.0 | 0.53 e ± 0.0 | 0.23 f ± 0.0 | 0.1 f ± 0.0 | 0.11 f ± 0.0 | 1.42 fg ± 0.0 | 1.72 bc ± 0.0 | 2.84 f ± 0.3 | 3.65 de ± 0.0 |
| 60–80 | 1.36 f ± 0.1 | 2.04 d ± 0.0 | 0.8 c ± 0.1 | 0.91 b ± 0.0 | 0.31 e ± 0.0 | 0.06 g ± 0.0 | 1.46 efg ± 0.0 | 1.8 ab ± 0.0 | 3.93 cd ± 0.1 | 4.81 b ± 0.0 |
| 80–100 | 2.27 c ± 0.0 | 1.7 e ± 0.2 | 1.02 a ± 0.0 | 0.57 e ± 0.0 | 0.03 h ± 0.0 | 0.06 g ± 0.0 | 1.55 de ± 0.0 | 1.87 a ± 0.0 | 4.87 b ± 0.1 | 4.2 c ± 0.0 |
| IP | * | | * | | * | | * | | * | |
| D | * | | * | | * | | * | | * | |
| IP × D | * | | * | | * | | * | | * | |

**(b) TWW**

| Irrigation period with TWW (IP) | $Ca^{2+}$ (mmoleL$^{-1}$) | | $Mg^{2+}$(mmoleL$^{-1}$) | | $K^+$(mmoleL$^{-1}$) | | $Na^+$(mmoleL$^{-1}$) | | CEC (mmoleL$^{-1}$) | |
|---|---|---|---|---|---|---|---|---|---|---|
| $P_1$ (38 years) | 2.2 | | 1.0 | | 0.6 | | 1.0 | | 4.7 | |
| $P_2$ (20 years) | 2.3 | | 1.0 | | 0.5 | | 1.2 | | 5.0 | |
| Depth (cm) (D) | $P_1$ | $P_2$ | $P_1$ | $P_2$ | $P_1$ | $P_2$ | $P_1$ | $P_2$ | $P_1$ | $P_2$ |
| 0–20 | 2.6 a ± 0.0 | 3.3 a ± 0.0 | 0.9 a ± 0.0 | 1.4 a ± 0.0 | 0.8 a ± 0.0 | 1.0 a ± 0.0 | 0.7 a ± 0.0 | 0.6 a ± 0.0 | 5.1 a ± 0.1 | 6.4 a ± 0.1 |
| 20–40 | 2.6 a ± 0.0 | 1.8 a ± 0.0 | 0.7 a ± 0.0 | 0.9 a ± 0.0 | 0.5 a ± 0.0 | 0.4 a ± 0.0 | 1.1 a ± 0.0 | 0.9 a ± 0.1 | 4.9 a ± 0.1 | 4.0 a ± 0.1 |
| 40–60 | 2.0 a ± 0.5 | 2.3 a ± 0.0 | 1.1 a ± 0.0 | 0.9 a ± 0.0 | 0.4 a ± 0.0 | 0.2 a ± 0.0 | 1.0 a ± 0.0 | 1.7 a ± 0.0 | 4.5 a ± 0.5 | 5.2 a ± 0.1 |
| 60–80 | 1.8 a ± 0.1 | 2.0 a ± 0.1 | 1.1 a ± 0.0 | 0.8 a ± 0.1 | 0.6 a ± 0.0 | 0.3 a ± 0.0 | 0.9 a ± 0.0 | 1.3 a ± 0.0 | 4.4 a ± 0.2 | 4.3 a ± 0.1 |
| 80–100 | 2.0 a ± 0.0 | 2.0 a ± 0.3 | 1.1 a ± 0.0 | 0.8 a ± 0.0 | 0.5 a ± 0.0 | 0.7 a ± 0.0 | 1.2 a ± 0.0 | 1.4 a ± 0.0 | 4.8 a ± 0.1 | 5.0 a ± 0.4 |
| IP | ns | | ns | | ns | | ns | | ns | |
| D | ns | | ns | | ns | | ns | | ns | |
| IP × D | ns | | ns | | ns | | ns | | ns | |

GW: groundwater; TWW: treated wastewater; CEC: cation exchange capacity; $P_1$: Oued Souhil area; $P_2$: BeniKhiar area; significant differences between different levels for each depth are indicated by lower case letters; ns: not significant; *: significant at $p < 0.05$.

As shown in Table 5, compared to GW, irrigation with TWW led to a significant increase in $Ca^{2+}$, $Mg^{2+}$, and $K^+$ soil concentrations and CEC. More interestingly, this increase particularly concerned the two elements $Ca^{2+}$ and $K^+$ at the top soil layer. Indeed, regardless of the duration of irrigation with TWW, the CEC in GW irrigated soils was 10% lower than CEC in TWW irrigated soils. The soil samples from plots irrigated with TWW had $Ca^{2+}$, $Mg^{2+}$, and $K^+$ contents that were 18, 71 and 68% higher, respectively, than soil samples from GW irrigated plots. $Na^+$ content in GW irrigated soils was 40% higher than in TWW irrigated soils. For both the studied areas ($P_1$ and $P_2$), when GW was used for irrigation, the concentration of $K^+$ decreased by 93% from the soil surface (0–20 cm) to the deepest layer (80–100 cm), while the concentration of $Mg^{2+}$ increased by 50% from the soil surface (0–20 cm) to the deepest layer (80–100 cm). Regardless of the water type and duration of TWW irrigation, the $Ca^{2+}$ content was higher in the 0–20 cm (2.8 mmoleL$^{-1}$) layer and in the deepest layer (2 mmoleL$^{-1}$) than in the medium soil layers (20–60 cm). In the same way, the soil concentrations of $K^+$ and $Mg^{2+}$ in TWW irrigated soils were greater in top soil ($K^+$ and $Mg^{2+}$ concentrations of 0.9 and 1.1 mmoleL$^{-1}$, respectively) and in the deepest layers ($K^+$ and $Mg^{2+}$ concentrations of 0.6 and 0.9 mmoleL$^{-1}$, respectively) than in the medium layers ($K^+$ and $Mg^{2+}$ concentrations of 0.4 and 0.9 mmoleL$^{-1}$, respectively).

### 3.4. Soil Adsorption Ratio (SAR)

Soil SAR was not significantly affected by the period of irrigation with TWW and did not significantly differ with depth in soils irrigated with TWW. However, using GW, the period of irrigation significantly affected the soil SAR ($p < 0.05$). The effects of water type and depth on soil sodicity were significant in both perimeters. In general, soil SAR was much higher in plots irrigated with GW (1.4 and 1.5 mmole$^{0.5}$ L$^{-0.5}$ in $P_1$ and $P_2$, respectively) than soils irrigated with TWW (0.8 and 0.9 mmole$^{0.5}$ L$^{-0.5}$ in $P_1$ and $P_2$, respectively) (Table 4).

### 3.5. Cation Ratio for Soil Structural Stability (CROSS)

Soil CROSS was not significantly affected by the different durations of TWW application and remained equal over the entire depth up to 1 m. However, the period of irrigation with GW significantly affected the soil CROSS ($p < 0.05$). The effects of water type and depth on the soil CROSS were significant for both perimeters. Generally, soil CROSS was much higher in plots irrigated with GW (1.7 mmole$^{0.5}$ L$^{-0.5}$ in $P_1$ and $P_2$) than soils irrigated with TWW (1.1 and 1.2 mmole$^{0.5}$ L$^{-0.5}$ in $P_1$ and $P_2$, respectively) (Table 4).

### 3.6. Relationship between CEC, SAR, CROSS, and Cation Concentrations

The correlation between CEC and cation concentrations was assessed in TWW and GW irrigated soils regardless of irrigation period. As shown in Figure 2, a positive correlation was obtained between CEC and $Ca^{2+}$, $Mg^{2+}$, and $K^+$ concentrations (r = 0.91 ***, r = 0.53 **, and r = 0.64 ***, respectively) in TWW irrigated soils. In the case of GW irrigated soils, a positive correlation was obtained between CEC and $Ca^{2+}$, $K^+$, and $Na^+$ concentrations (r = 0.97 ***, r = 0.66 ***, and r = 0.11 $^{ns}$, respectively).

The treatment TWW led to a significant negative correlation between SAR and $Ca^{2+}$ and $Mg^{2+}$ (r = −0.46 ** and r = −0.6 ***, respectively) and between CROSS and $Ca^{2+}$ and $Mg^{2+}$ (r = −0.45 * and r = −0.52 **, respectively) (Figure 3). GW application showed a significant negative correlation between $Mg^{2+}$ and $K^+$ concentrations (r = −0.54 **) in both the $P_1$ and $P_2$ study areas (Figure 4).

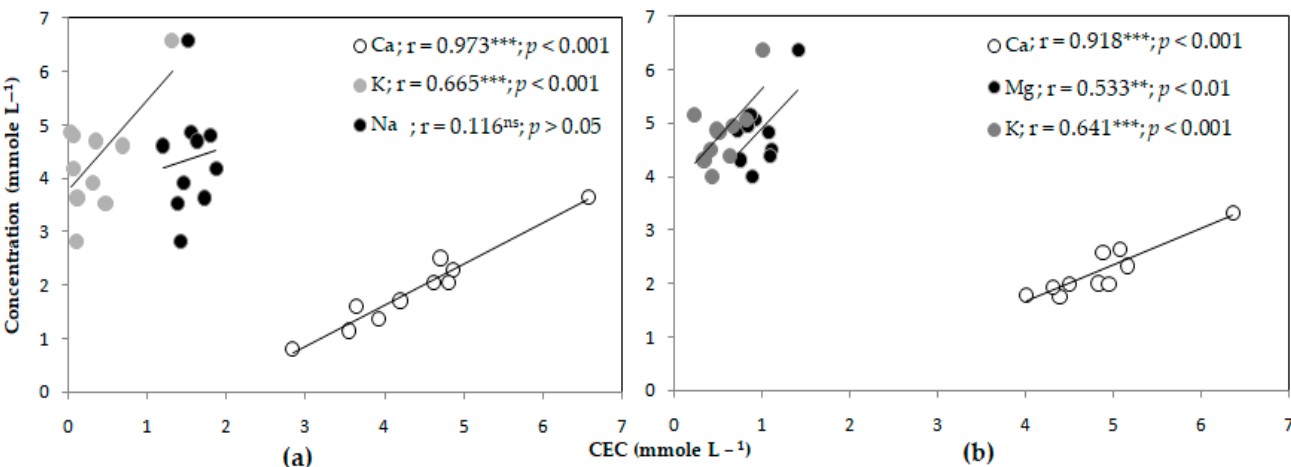

**Figure 2.** Linear regressions showing the relationship between calcium ($Ca^{2+}$), magnesium ($Mg^{2+}$), potassium ($K^+$), and sodium ($Na^+$) concentrations for cation exchange capacity (CEC) in soils irrigated with (**a**) groundwater and (**b**) treated wastewater. Cation concentrations and CEC are averaged across all depths. **, *** Significant at $p < 0.01$, and $p < 0.001$, respectively. ns = Not significant.

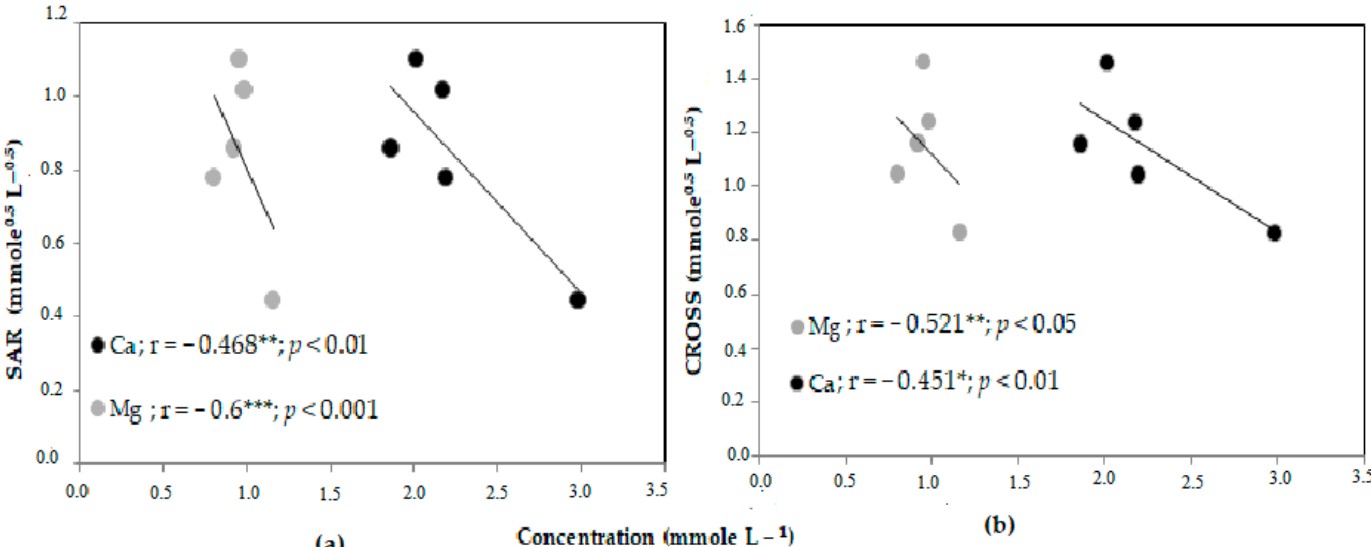

**Figure 3.** Linear regressions showing the relationship between calcium ($Ca^{2+}$) and magnesium ($Mg^{2+}$) concentrations for (**a**) sodium adsorption ratio (SAR) and (**b**) cation ratio of structural stability (CROSS). Cation concentrations, SAR, and CROSS are averaged across soils irrigated with treated wastewater and across all depths. *, **, *** Significant at $p < 0.05$, $p < 0.01$, and $p < 0.001$, respectively.

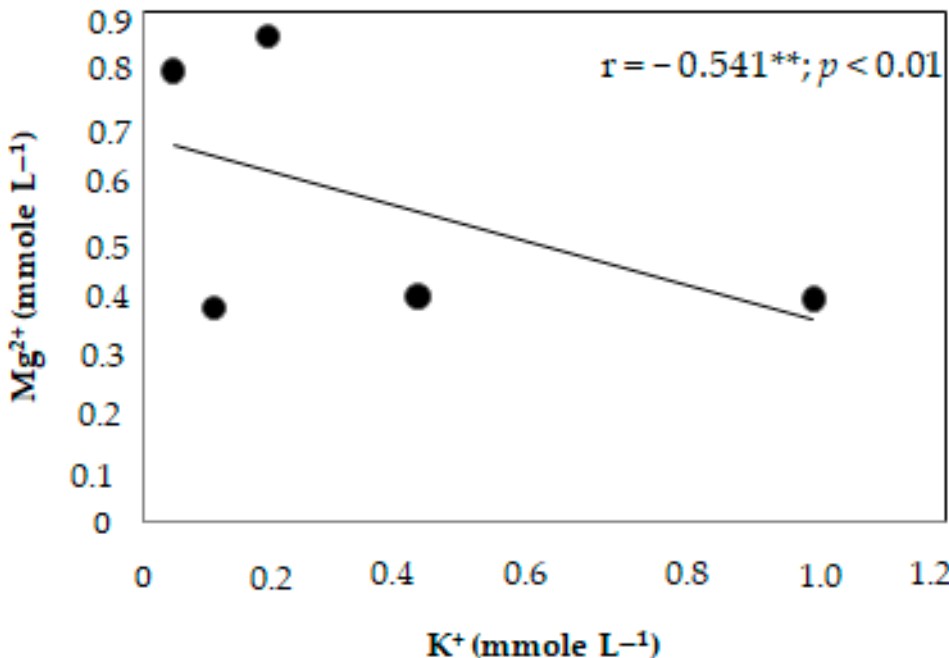

**Figure 4.** Linear regression showing the relationship between magnesium ($Mg^{2+}$) and potassium ($K^+$) concentrations in groundwater irrigated soils. $Mg^{2+}$ and $K^+$ concentrations are across the two areas of study and averaged across all five depths. ** Significant at $p < 0.01$.

## 4. Discussion

### 4.1. Effects of TWW and Application Period on Physicochemical Properties

Irrigation with treated wastewater significantly altered all the physicochemical parameters. In the current study, the TWW period only had an impact on the soil pH. Indeed, compared to irrigation period of 20 years, soil pH increased after 38 years of wastewater application. This increase might be ascribed to processes such denitrification [21], which requires one mole of $H^+$ for every mole of denitrified $NO_3^-$ and decarboxylation and deamination (organic anions and amino acids) activities that use protons [22]. Previous studies [23,24] have also reported increased soil pH with wastewater irrigation. In addition, according to Hidri et al. [25], the increase in pH after long-term TWW application could be explained by the modification in the bacterial community structure as a result of a decrease in abundance of Actinobacteria and an increase in abundance of Gammaproteobacteria, which is positively correlated to soil pH [26]. In contrast to TWW, the long-term application of GW had altered all soil parameters except the pH. In fact, we noted a decrease in EC, CEC, SAR, and CROSS properties. The improvement of soil traits after 38 years of irrigation with GW might be attributed to the increase in $Mg^{2+}$ concentration and the decrease in $K^+$ and $Na^+$ contents in the soil solution. Indeed, the continuous application of GW can cause the accumulation of $Mg^{2+}$ cation, which enhances the leaching of $K^+$ and $Na^+$ cations to the deep soil layers [27].

The results of this study showed that soil EC was not significantly influenced by the duration of applied TWW; it was only affected by the water type. Similarly, Barbosa et al. [28] analyzed the physicochemical parameters in soil irrigated with TWW and also found that long-term wastewater application did not increase the soil salinity. In contrast, Khaskhoussy et al. [29] used wastewater in irrigation and observed an increase in soil salinity. In fact, the richness of organic matter in TWW may result in an increase in nutrients that affect the EC of the soil [24]. Higher soil salinity associated with TWW irrigation could be due to the higher amounts of soluble salts in TWW than in GW, which may cause soil salinization.

The composition of cations in the soil solution and the CEC were not influenced by the different durations of TWW application in the $P_1$ and $P_2$ study areas. However, the different

water types did affect the two parameters. The CEC is directly related to the capacity of soil to absorb or exchange cations [30]. The CEC values obtained in this study were higher in the soil treated with TWW than GW. These obtained values can be attributed to the relatively high soil organic matter (SOM)in wastewater [31]. Opoku-Kwanowaa et al. [32] stated that organic matter is responsible for the CEC of soils. Because organic matter is negatively charged, the more negatively charged the surfaces that are present and available, the more positively charged ions (cations) they will attract. The CEC difference between GW irrigated soils and TWW irrigated soils corresponded with higher concentrations of $Ca^{2+}$, $Mg^{2+}$, and $K^+$ in TWW than GW. The positive correlation between CEC and $Ca^{2+}$, $Mg^{2+}$, and $K^+$ concentrations of TWW irrigated soils indicated that CEC of TWW irrigated soils was occupied by $Ca^{2+}$, $Mg^{2+}$, and $K^+$ in order of importance. The increase in $Ca^{2+}$, $Mg^{2+}$, and $K^+$ concentrations due to wastewater application agrees with the findings of Tarchouna et al. [33] and Heidarpour et al. [34], who also reported an increase in these elements after irrigation with wastewater. On the other hand, the positive correlation between CEC and $Ca^{2+}$, $Na^+$, and $K^+$ concentrations of GW irrigated soils showed that CEC of GW irrigated soils was occupied by $Ca^{2+}$, $Na^+$, and $K^+$ in order of importance. Indeed, soils irrigated with GW are characterized by much larger sodium content than soils irrigated with TWW.

The current study demonstrated that in terms of sodium adsorption ratio (SAR), all values were below the FAO safe limits (3.0). However, we noted that the SAR of soil irrigated with TWW was lower than that with GW. Contrarily to our findings, a set of studies reported that the SAR was higher for TWW than GW [35]. In our case, these findings might be explained by the natural accumulation of $Na^+$ ion concentration in soil solution in the case of GW. In addition, according to Changati et al. [36], TWW is relative to divalent cations that facilitate $Na^+$ removal and its leaching into deeper soil layers.

The data also revealed that the CROSS was higher for GW than TWW, indicating that the latter aided in maintaining soil structure. The results provide additional and convincing evidence in support of earlier arguments presented in the literature that TWW is a potential option to improve the soil structure [9]. The significant negative relationship between CROSS and both $Mg^{2+}$ and $Ca^{2+}$ further support the beneficial effect of TWW in improving soil structural stability. More interestingly, these results indicate that the decrease in CROSS in soils irrigated with TWW is relative to divalent cations present in TWW, which facilitate $Na^+$ removal and its leaching into deeper soil layers. Thus, TWW might be considered as a good solution to reclaim soil sodicity and structural stability as it readily supplies $Ca^{2+}$ and $Mg^{2+}$ cations to counter $Na^+$ cation in the soil solution [27].

*4.2. Effects of the Interactive Soil Layer and Water Type on Physicochemical Attributes*

Our data revealed that at both perimeters, the migration of salt was governed by sandy soil texture, lateral drainage flows, and principally water irrigation composition, which resulted in a salt accumulation trend in the subsoil (60–100 cm) for TWW but only in the soil surface (0–40 cm) in the case of GW. The trend for TWW might be attributed to leaching and drainage, with the richness of divalent cations in TWW causing leaching of monovalent cations (e.g., $K^+$ and $Na^+$) in the deeper subsoil. In addition, this trend might be explained by the improvement of permeability due to TWW irrigation, as mentioned by Azouzi et al. [1]. More importantly, the specific trend of the $K^+$ cation, whichhad increased concentration in the soil surface and the deeper subsoil, indicates a likely fertilizing effect of TWW. This was proven in a previous study conducted on an experimental field irrigated with treated wastewater [37]. The highest amount of salt in the upper depth might be attributed to the arid climate of this region, which is characterized by high temperatures and low precipitation. This is in accordance with the findings of Liu et al. [38]. In soils irrigated with GW, the high $K^+$ cation values in the upper soil layers indicated that this element did not move rapidly through the profile of the soil as a consequence of the equilibrium reached between the element in solution and that adsorbed on soil colloids [39]. The lower $K^+$ concentration in the bottom soil layers was probably a consequence of plant uptake or

movement of $K^+$ from the soil solution to plant tissues [35]. On the other hand, there was highly significant accumulation of $Mg^{2+}$ in the deepest layer (80–10cm) in soils irrigated with GW and lower $Mg^{2+}$ content in the top soil layers (0–20 cm). This can be explained by the antagonistic activity of $K^+$, which reduces the adsorption of $Mg^{2+}$ on exchangeable complexes and enhances the leaching of $Mg^{2+}$ to the deepest soil layers [40]. The negative correlation between the concentrations of $K^+$ and $Mg^{2+}$ in GW irrigated soils confirms this hypothesis (Figure 4). Regarding the behavior of $Ca^{2+}$, regardless of the water irrigation type, the abundance of this element in the upper soil was principally associated with its negative effect on $Na^+$ movement from top soil to lower depths (60–100 cm). In particular, the richness of organic matter in TWW promoted $CaCO_3$ dissociation to liberate more calcium and consequently reduce the soil sodicity [41]. These results are in qualitative agreement with the results of Belaid et al. [42], who found that irrigation with TWW clearly increased the leaching of $Na^+$ cation.

## 5. Conclusions

To the best our knowledge, the effect of TWW application period compared to GW in sandy soil has been little explored. This research showed that although TWW application increased the salinity, it improved other physicochemical attributes, particularly structural stability, sodicity, and richness of nutrients in the soil (e.g., $Ca^{2+}$, $Mg^{2+}$, and $K^+$). Consequently, TWW promoted the leaching of salt cations outside the root zone. Additionally, in the present study, the TWW application period only had an impact on soil pH, whereas GW had an impact on all parameters with the exception of soil pH. Although the soil quality did improve after 38 years of GW treatment, the improvement in soil properties through long-term application of TWW was more significant. This report underlines the positive effect of TWW even after long-term application, thus showing the importance of soil texture for farmers applying TWW. However, these findings only relate to sandy field, and the validity of our conclusions need to be verified for other types of soils irrigated with TWW.

**Author Contributions:** Conceptualization, S.B.; methodology, S.B., R.I.Z., M.N.K. and H.B.; software, S.B.; validation, H.B. and M.M.; formal analysis, S.B., O.N., E.T. and R.G.; investigation, S.B. and M.N.K.; resources, M.N.K.; data curation, S.B.; writing—original draft preparation, S.B.; writing—review and editing, K.B.; visualization, R.I.Z.; supervision, M.M.; project administration, H.B.; funding acquisition, M.M. All authors have read and agreed to the published version of the manuscript.

**Funding:** This research was funded by the Arid Region Institute of Tunisia.

**Institutional Review Board Statement:** Not applicable.

**Informed Consent Statement:** Not applicable.

**Data Availability Statement:** Due to the confidentiality agreement requirements data is not available.

**Conflicts of Interest:** The authors declare no conflict of interest.

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
