# Peer review of "Soil Physicochemical Changes as Modulated by Treated Wastewater after Medium-and Long-Term Irrigations: A Case Study from Tunisia"

_agriculture, doi:10.3390/agriculture12122139_

Round 1

Reviewer 1 Report

I have read the manuscript. It work evaluates the effect of irrigation on physiochemical properties of arid soils. There are some concerns with the work.

1. Grammar needs to be worked on seriously.

2. The title is confusing. The title should reflect the effect of irrigation on physicochemical parameters.

3. Line 38. write hectares in words. 

4. Check the symbol for pH and spelling of standard in Table 2.

5. How about comparison with regions that are not exposed to TWW irrigation? 

Author Response

Dear reviewer,

 First, we would like to thank you for your constructive and encouraging feedback which improved our manuscript (agriculture-2000730) considerably. We made all the revisions according to reviewer concerns. Please find the answers for each of the points raised by the reviewer in the present MS word file. Please refer to the responses as follows: reviewer comments in normal font, author’s response in blue, changes in the MS in red.

Reviewer #1:

*Comment 1: I have read the manuscript. It work evaluates the effect of irrigation on physiochemical properties of arid soils. There are some concerns with the work.

  1. Grammar needs to be worked on seriously.

Reply 1: Thank you for your comment, as suggested; we have checked grammar in the whole of manuscript.

*Comment 2: The title is confusing. The title should reflect the effect of irrigation on physicochemical parameters.

Reply 2: We thank the reviewer for pointing this out and we apologize for the inaccuracy. The title has been modified as suggested, giving an idea of the experiment and the major obtained results.

Soil physico-chemical traits as modulated by treated wastewater after medium and long-term irrigations: A study case from Tunisia” (L 2-4).

*Comment3: Line 38. write hectares in words

Reply 3: As requested, the abbreviation “ha” was written in words “hectars” (L46; L53).

*Comment4: Check the symbol for pH and spelling of standard in Table 2.

Reply 4: Sorry for this omission, the term “Ph” in table 3 was changed by “pH” (L132). The spelling mentioned in the table 2 was explained as footnote above the table 2 (L133).

*Comment5: How about comparison with regions that are not exposed to TWW irrigation? 

Reply 5: As recommended, we added the comparison between regions exposed to GW in the two sections: results and discussion (L180-181; L192-193; L199-203; L229-230; L237-238; L279-285).  

Reviewer 2 Report

The manuscript titled “Soil chemical traits as modulated by treated wastewater after medium and long-term irrigations: A study case from Tunisia”, explored the medium and long-term effects of TWW on soil physico-chemical parameters.

This is an interesting paper, which addresses an important issue with regards of the wastewater reuse. However, it needs some major editing.

There are a lot of grammar or typing mistakes (some of them are marked in review below). Also, the results provided in Tables duplicates the ones in Figures (the duplicate results must be excluded). Also, Table 3 includes some data with commas instead of dots.

 Abstract. Needs some sentence, on highlighting the importance of the obtained results or what problems could be solved, etc.

Keywords: “treated wastewater; …” – must differ from the title. Suggest, to change it to, for example, “water reuse” or other words.

 Lines 82–85: “To contribute to the current understanding of soil fertility, two perimeters differed by the TWW lenght application were studied in sandy soil texture. Differents chemicals soil properties and sodicity and structural statbility in- dexes were investigated in North-Est of Tunisia” –The aim of the study must be highlighted.

 Lines 117–118: “The soil texture at the two perimeters is sandy with 79% sand, 7.5% silt and 13.5% clay” – How it is possible, that in two perimeters, soil texture has exactly the same amount of sand, silt and clay? Please indicate the standard deviation as well as indicate the soil texture for each investigated depth, because soil texture usually differs between the depths too.

 Lines 126 “For each replication, five soil cores were collected and homogeinized.” – please indicate how many soil cores were collected per each depth, because five soil cores per replicate, looks like you took one soil core per each depth, per 3 replications, this means, that you only had 3 soil cores per each depth. Please clarify.

 Lines 261, 262 “…soil pH decreased after 20 years application of wastewater and increased after 38 years.” – For this statement you need whether to indicate the pH at the beginning of the study (20 and 38 years ago), or you need to have a control without the irrigation with wastewater (for example irrigated with clean water). Otherwise, you can only compare differences between 20 and 38 years of irrigation.

 Conclusions should be more concrete and respond the obtained results.

Author Response

Dear reviewer,

 First, we would like to thank you for your constructive and encouraging feedback which improved our manuscript (agriculture-2000730) considerably. We made all the revisions according to reviewer concerns. Please find the answers for each of the points raised by the reviewer in the present MS word file. Please refer to the responses as follows: reviewer comments in normal font, author’s response in blue, changes in the MS in red.

Reviewer #2:

General comments: The manuscript titled “Soil chemical traits as modulated by treated wastewater after medium and long-term irrigations: A study case from Tunisia”, explored the medium and long-term effects of TWW on soil physico-chemical parameters.

This is an interesting paper, which addresses an important issue with regards of the wastewater reuse. However, it needs some major editing.

Reply: Thank you for your positive feedback and constitutive remarks; we followed your comments mentioned bellow point by point.

*Comment1: There are a lot of grammar or typing mistakes (some of them are marked in review below).

Reply 1: Sorry for this grammar mistakes, we checked the whole of the manuscript to avoid any typing errors and improve the English.  

*Comment2: Also, the results provided in Tables duplicates the ones in Figures (the duplicate results must be excluded)

Reply 2: We thank the reviewer for his comment. We agree with you that to avoid redundancy of information contained in the tables and figures, we have kept only the tables. We have modified the manuscript accordingly.

*Comment3: Also, Table 3 includes some data with commas instead of dots.

Reply 3: Sorry for this omission, we checked all tables, deleted commas and replace it with dots.

*Comment4: Abstract. Needs some sentence, on highlighting the importance of the obtained results or what problems could be solved, etc.

Reply 4: Your point of view has been considered. The abstract was improved to mention the problematic and highlighting the importance of the obtained results given that we have been limited by a number of words (200 words) (L34-36).

*Comment5: Keywords: “treated wastewater; …” – must differ from the title. Suggest, to change it to, for example, “water reuse” or other words.

Reply 5: As suggested, the term “treated wastewater” in the keywords was changed by “water reuse” (L37).

*Comment6: Lines 82–85: “To contribute to the current understanding of soil fertility, two perimeters differed by the TWW lenght application were studied in sandy soil texture. Differents chemicals soil properties and sodicity and structural statbility in- dexes were investigated in North-Est of Tunisia” –The aim of the study must be highlighted.

Reply 6: As requested, the aim of this investigation was highlighted in the end of the introduction section (L95-99).

*Comment 7: Lines 117–118: “The soil texture at the two perimeters is sandy with 79% sand, 7.5% silt and 13.5% clay” – How it is possible, that in two perimeters, soil texture has exactly the same amount of sand, silt and clay? Please indicate the standard deviation as well as indicate the soil texture for each investigated depth, because soil texture usually differs between the depths too.

Reply 7: Sorry for this omission. In fact, we added a table 3 that mention the details of the soil texture according to soil depth. Also, for each amount of sand, silt and clay we indicated the standard deviation in the average value for the two perimeters (L134-138). 

*Comment 8: Lines 126 “For each replication, five soil cores were collected and homogeinized.” – please indicate how many soil cores were collected per each depth, because five soil cores per replicate, looks like you took one soil core per each depth, per 3 replications, this means, that you only had 3 soil cores per each depth. Please clarify.

Reply 8: Based on your comment, in order to further clarify the soil sampling method, we have added some information to detail more the different steps adopted to collect the soil from each studied perimeter. The sentence was modified as “At each irrigated perimeter, three plots were selected for each irrigation source (TWW and GW). For each plot, sampling was carried out by taking soil from the five depth layers. In each soil depth, five cores were sampled and homogenized. Samples were collected using an auger for a total of thirty samples for each perimeter” (L143-146).

*Comment9: Lines 261, 262 “…soil pH decreased after 20 years application of wastewater and increased after 38 years.” – For this statement you need whether to indicate the pH at the beginning of the study (20 and 38 years ago), or you need to have a control without the irrigation with wastewater (for example irrigated with clean water). Otherwise, you can only compare differences between 20 and 38 years of irrigation.

Reply 9: As recommended, to ovoid the confusing about the effect of period of TWW application on pH value, an additional information were added in the beginning of the discussion section (L270-279).

*Comment10: Conclusions should be more concrete and respond the obtained results.

Reply 10: Your point of view has been considered, the conclusion section was modified giving more details about obtained results in this investigation (L372-376).

Round 2

Reviewer 1 Report

Most of my concerns have been addressed. The authors should consider changing the title to "Soil physico-chemical changes as modulated by treated 2 wastewater after medium and long-term irrigations: A study 3 case from Tunisia"

Reviewer 2 Report

Dear authors,

thank you for the updated manuscript according to suggestions. The improved version of the manuscript looks now suitable for publishing.